# Low Sintering Temperature Nano-Silver Pastes with High Bonding Strength by Adding Silver 2-Ethylhexanoate

**DOI:** 10.3390/ma14205941

**Published:** 2021-10-10

**Authors:** Steve Lien-Chung Hsu, Yen-Ting Chen, Meng-Liang Chen, In-Gann Chen

**Affiliations:** Department of Materials Science and Engineering, National Cheng-Kung University, Tainan 701-01, Taiwan; ytc8741@gmail.com (Y.-T.C.); qqliang007@gmail.com (M.-L.C.); ingann@mail.ncku.edu.tw (I.-G.C.)

**Keywords:** silver nanoparticles, silver precursor, nano-silver pastes, copper-to-copper bonding, low temperature sintering

## Abstract

A silver precursor (silver 2-ethylhexanoate) and silver nanoparticles were synthesized and used to prepare a low sintering temperature nano-silver paste (PM03). We optimized the amount of silver 2-ethylhexanoate added and the sintering temperature to obtain the best performance of the nano-silver paste. The relationship between the microstructures and properties of the paste was studied. The addition of silver 2-ethylhexanoate resulted in less porosity, leading to lower resistivity and higher shear strength. Thermal compression of the paste PM03 at 250 °C with 10 MPa pressure for 30 min was found to be the proper condition for copper-to-copper bonding. The resistivity was (3.50 ± 0.02) × 10^−7^ Ω∙m, and the shear strength was 57.48 MPa.

## 1. Introduction

To improve operation speed, it is necessary to increase the density of transistors in a device. At present, the line width of electronic devices is hitting a physical limitation. To increase the density of transistors, a three-dimensional integrated circuit (3D IC) package is developed [1,2,3]. For integrating different IC chips, the chips have to be connected. Therefore, the junctions are extremely important [4,5,6].

Usually, the adhesive for the 3D IC junction can be conventional tin–lead solder, lead-free solder, or a silver paste. The silver paste has higher bond strength, better conductivity, and excellent high temperature service performance compared to the conventional solders [7,8,9]. Due to these reasons, silver paste is a good adhesive candidate for reducing current consumption and lowering heat caused by resistance [10,11,12]. By choosing an appropriate dispensing method, the silver paste can create a junction with a diameter less than 10 microns [13]. Compared to a solder ball with a diameter of 100 microns, the silver paste has greater potential for use in wide bandgap technologies and die attach applications [14,15].

The surface-to-volume ratio of nanoparticles is so large that they can be sintered at lower temperatures than bulk materials. Silver nanoparticles are used to prepare conductive inks and pastes due to this property [16,17,18]. With a through-silicon via (TSV) connection, nano-silver pastes can bond chips together via copper-to-copper bonding. These bonds are created by sintering individual silver nanoparticles together onto copper surfaces at low temperatures [19]. Therefore, nano-silver pastes can be used for 3D IC integration [20,21].

In our previous study, we used a silver precursor, silver 2-ethylhexanoate, added to polyimide to form conductive polyimide nanocomposites [22]. The silver 2-ethylhexanoate can be converted into silver via an in situ reduction at a high temperature. In this study, we combined silver 2-ethylhexanoate with a nano-silver paste to improve the resistivity and shear strength of the paste. The name of the paste was coded as PM03. The relationship between the addition of different amounts of silver 2-ethylhexanoate and changes in sintering temperature was studied to obtain the proper composition and process conditions.

## 2. Experimental

### 2.1. Materials

Silver nitrate was obtained from UniRegion Bio-tech. The chemicals 1-propylamine, 2-ethylhexanoic acid, and heptanoic acid were purchased from Alfa Aesar, Sodium borohydride was obtained from Acros. Sodium hydroxide was obtained from Showa. Dimethylacetamide (DMAc) was purchased from Tedia.

### 2.2. Synthesis of Silver Nanoparticles

An amount of 500 mL of toluene and 14.78 g (0.25 mole) of 1-propylamine were put into a beaker. After stirring with a glass rod, the solution was put into a three-necked flask and stirred continuously with a mechanical stirrer. Precisely 21.23 g (0.125 mole) of silver nitrate was added into the solution, and 94.12 g (0.723 mole) of heptanoic acid was added after all of the silver nitrate was dissolved. The solution then changed from transparent to white. After 15 min, 2.36 g (0.0625 mole) of sodium borohydride was added, and the color of the solution changed from white to black. After reacting for 1 h, the sample was washed with an acetone and methanol mixture and filtrated 3 times. After being dried in vacuum for 12 h, the dark blue silver nanoparticles were thus obtained.

### 2.3. Synthesis of Silver 2-Ethylhexanoate (CH_3_(CH_2_)_3_CH(C_2_H_5_)CO_2_Ag)

Precisely 2.32 g (0.058 mole) of sodium hydroxide and 8.36 g (0.058 mole) of 2-ethylhexanoic acid were dissolved in 50 mL of DI water and 50 mL of methanol, respectively. The two solutions were mixed. The resulting solution was labeled as solution A. An amount of 9.85 g (0.058 mole) of silver nitrate was dissolved in 50 mL of DI water, and this solution was labeled as solution B. Solution B was then added into solution A dropwise. The sample was washed with methanol and filtrated 2 times in order to obtain the white precipitate. The precipitate was collected by filtration, washed with distilled water and methanol, and then dried in vacuum. The silver 2-ethylhexanoate was then obtained.

### 2.4. Preparation of Paste PM03

The silver nanoparticles, silver 2-ethylhexanoate, and DMAc were mixed and stirred with a glass rod in a beaker according to the compositions depicted in Table 1. The mixture was uniformly mixed with a three-roller mixing grinder to form a nano-silver paste (PM03).

### 2.5. Characterization

The X-ray diffraction (XRD) analysis of silver nanoparticles and silver precursor was conducted on a Rigaku D/MAX-IIIV X-ray Diffractometer using Ni-filtered Cu-Kα radiation with a scanning rate of 4°/min^−1^ at 30 kV and 20 mA.The textures of the pastes were investigated by a Hitachi Stereoscan 260 scanning electron microscope (SEM) at a 10 kV operating voltage.Fourier transform infrared spectroscopy (FTIR) analysis of silver 2-ethylhexanoate was performed on a JASCO FT/IR 4600 spectrometer with a KBr pellet.The weight losses of the silver nanoparticles and silver 2-ethylhexanoate were analyzed using a TA Instrument Thermogravimetric Analyzer (TGA) 2050 at a heating rate of 10 °C/min under air.The resistivity measurement was conducted as follows: A glass substrate was cleaned with ethanol. The two sides of the glass substrate were taped with two layers of 3M Scotch 810 tape. The paste PM03 was transferred to the glass substrate, and the thickness of the paste was fixed with blade coating. The 3M Scotch 810 tape was removed, and the sample was heated. The sheet resistance and thickness were measured with 4-point probes and a thickness gauge, respectively, in order to calculate the resistivity of the paste.The shear strength measurement was conducted as follows: The surfaces of the polished copper blocks were cleaned with acetone, 25 wt% citric acid solution, DI water, and ethanol. The two sides of the copper blocks were taped with two layers of 3M Scotch 810 tape. Paste PM03 was transferred to the copper blocks and the thickness of the paste was fixed with blade coating. The 3M Scotch 810 tape was removed. Two copper blocks were put together and bonded using thermal compression. Then, the shear strength was measured using a shear strength testing machine. The thermal compression and shear strength test of the sintering Ag paste are shown in Figure 1.

## 3. Results and Discussion

### 3.1. Synthesis of Silver Nanoparticles

In our previous study, silver nanoparticles had been prepared through the reduction reaction of AgNO_3_ by formaldehyde with a catalyst [23]. In that study, poly(N-vinyl-2-pyrrolidone) (PVP) had been chosen as the protecting agent. Due to the high decomposition temperature of PVP, the sintering temperature of the silver nanoparticles was high. In this study, a low molecular weight organic compound, heptanoic acid, was used as the protecting agent to prepare the nanoparticles in order to reduce the sintering temperature. When we used sodium borohydride as the reductant and heptanoic acid as the protecting agent, the silver nanoparticles were successfully prepared from the AgNO_3_ precursor. The XRD pattern is shown in Figure 2a. The reflection peaks are indexed as the fcc (111), (200), (220), and (310) planes, of which the nanoparticles are silver [24]. The particle sizes were around 15–40 nm as shown in the TEM micrograph (Figure 2b). The TGA thermograph of silver nanoparticles is shown in Figure 3. As we can see from the graph, the decomposition of the protecting agent (heptanoic acid) starts from 180 °C and can be fully removed after 250 °C. It shows that heptanoic acid occupies 6.4% of the whole composition’s weight. If the sintering temperature is equal to or higher than 250 °C, the silver nanoparticles will lose protecting groups and be sintered by the driving energy of the temperature. Thus, a bulk material with excellent strength and conductivity is obtained.

### 3.2. Synthesis of Silver 2-Ethylhexanoate

The silver 2-ethylhexanoate was synthesized from the reaction of 2-ethylhexanoic acid and silver nitrate. 2-Ethylhexanoic acid solution in methanol was mixed with an aqueous solution of sodium hydroxide to form a mixed solution. Then, a solution of silver nitrate was added to the mixed solution to form a silver precursor. The product was characterized by FTIR as shown in Figure 4a. The absorption peak of the carboxylate anion in the silver 2-ethylhexanoate appeared between 1610–1550 cm^−1^. The absorption peak of the alkyl group was observed at around 2920 cm^−1^. Compared to the FTIR spectrum of 2-ethylhexanoic acid (Figure 4b), we can see that the absorption peak of the carbonyl group (1705 cm^−1^) in 2-ethylhexanoic acid shifted to 1550 cm^−1^ when it was converted to silver 2-ethylhexanoate. That proves the formation of the silver salt. From TGA analysis (Figure 5), the decomposition temperature of silver 2-ethylhexanoate started at 150 °C, and the maximum decomposition rate was at 183 °C. After decomposing completely, a 56.7% thermogravimetric loss is formed, which is in full compliance with the ratio of the 2-ethylhexyl acid group and silver in 2-ethylhexanoate. After heating at 250 °C for 30 min, the silver 2-ethylhexanoate was converted to silver as shown in the XRD pattern (Figure 6).

### 3.3. Effect of Silver 2-Ethylhexanoate Added to the Paste

The novelty of this study is the addition of silver 2-ethylhexanoate to the silver nanoparticle paste. When an appropriate amount of silver 2-ethylhexanoate is added to the silver nanoparticle paste, the in situ reduction at a high temperature helps to connect the sintered silver. This fills the voids in the sintered silver to reduce resistance and increase density and intensification. That makes the paste with low resistivity and high bonding strength at low temperature sintering.

The results of the resistivity measurements are shown in Table 2. The resistivity of the pastes sintered at 250 °C for 30 min decreased as the amount of silver 2-ethylhexanoate increased up to a 17.5 wt%. The resistivity of paste PM03-25 was higher than the resistivity of paste PM03-17.5. PM03-17.5 showed a low resistivity of (3.50 ± 0.02) × 10^−7^ Ω∙m. When the resistivity is reduced, we can decrease the heat generated by the electric current to save energy and solve the heat dissipation problem. Moreover, the resistivity is a factor of resistance–capacitance time delay [25]. The signal delay decreases as the resistance reduces, allowing the components to achieve further efficient performance. Therefore, the resistivity of the material, which is used as a junction adhesive, is extremely important [26].

The results of the shear strength test showed a similar behavior. As shown in Table 3, the shear strength increased as the amount of silver 2-ethylhexanoate was raised up to 17.5 wt%. The SEM images of the fracture surfaces with different amounts of silver 2-ethylhexanoate are shown in Figure 7. For PM03-0, the original boundary of the silver nanoparticles does not exist anymore due to the sintering. What is seen is a continuous sintered body, but with voids. The sintering between particles became denser as more silver 2-ethylhexanoate was added. From the data, we can see that, by adding the silver precursor, the strength of the junction of the nano-silver paste is significantly improved to obtain a high strength nano-silver paste. However, in the case of 25 wt% addition, large particles were formed due to the aggregation of silver 2-ethylhexanoate. The sintering between these large particles was worse than fine particles. The change in the microstructure of the paste affects the properties.

The Paste PM03-17.5 has shear strength as high as 57.48 MPa, which is suitable for use in 3D IC junctions. The junction needs to have high strength to avoid the breakdown of the copper blocks and render the failure of the junction. Therefore, the strength of the junction could influence the life of the product. Higher strength and better reliability give the product a longer life [27].

### 3.4. Effect of Sintering Temperature to the Paste

Sintering temperature is always an important parameter of the sintering process. The high sintering temperature will provide more energy to obtain a dense structure [28]. However, the silver paste developed in this research is for the junction of IC chips. The sintering temperature must not be too high to avoid damaging the IC chips [29,30]. Therefore, it is critical to find the lowest sintering temperature that can still achieve good sintering properties.

The results of the resistivity of different sintering temperatures are shown in Table 4. As the sintering temperature increased, the resistivity decreased. In Table 5, it can be seen that the shear strength increased as the sintering temperature increased. The SEM images of the sintered paste in Figure 8 showed that higher sintering temperatures led to better sintering and denser microstructures, thus resulting in better properties. At 200 °C, the silver nanoparticles did not have good sintering. They formed several clusters, but not a continuous film structure. There was little difference in terms of improvement found in the properties of paste PM03-17.5 sintered at 300 °C compared to 250 °C. Therefore, sintering at 250 °C was determined to be high enough for paste PM03-17.5. The comparison of PM03-17.5 with other the state-of-the-art silver pastes is shown in Table 6.

## 4. Conclusions

Silver nanoparticles and silver 2-ethylhexanoate were successfully synthesized, and these two components were combined to produce a nano-silver paste: paste PM03. Adding silver 2-ethylhexanoate resulted in better sintering for the paste, but too much silver 2-ethylhexanoate led to larger particles and worse sintering. Higher heating temperature led to lower resistivity and higher shear strength due to better sintering. However, sintering at temperatures of 250 °C and 300 °C resulted in little difference. Thermal compression at 250 °C for 30 min with 10 MPa pressure with paste PM03-17.5 was found to be the optimal parameter. The resistivity was (3.50 ± 0.02) × 10^−7^ Ω∙m, and the shear strength was 57.48 MPa.

## Figures and Tables

**Figure 1 materials-14-05941-f001:**
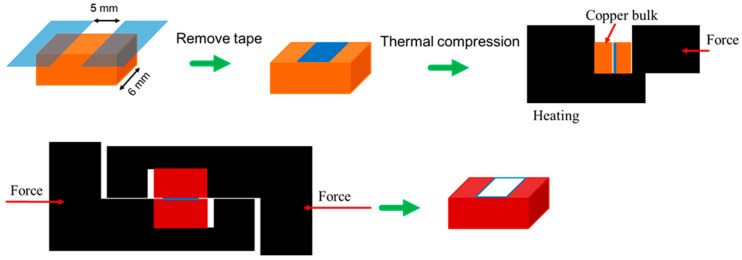
Thermal compression and shear strength test of sintering Ag paste.

**Figure 2 materials-14-05941-f002:**
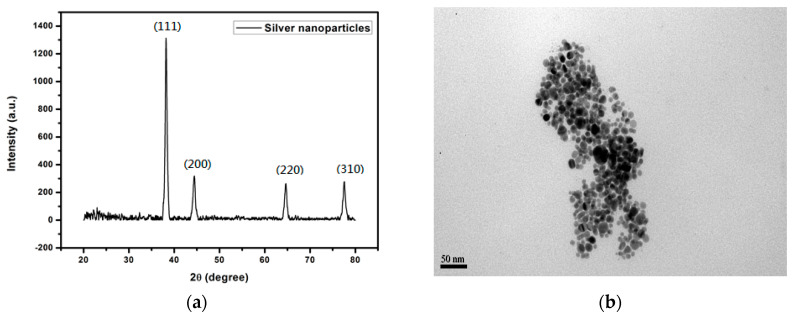
XRD pattern (**a**) and TEM micrograph (**b**) of silver nanoparticles.

**Figure 3 materials-14-05941-f003:**
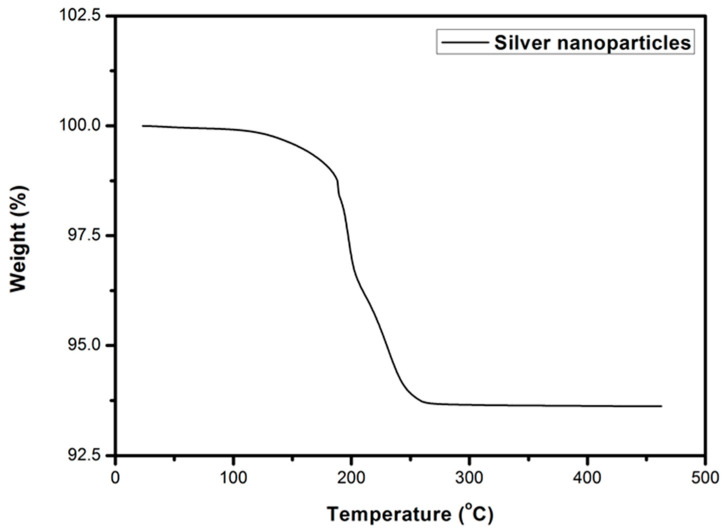
TGA thermograph of silver nanoparticles.

**Figure 4 materials-14-05941-f004:**
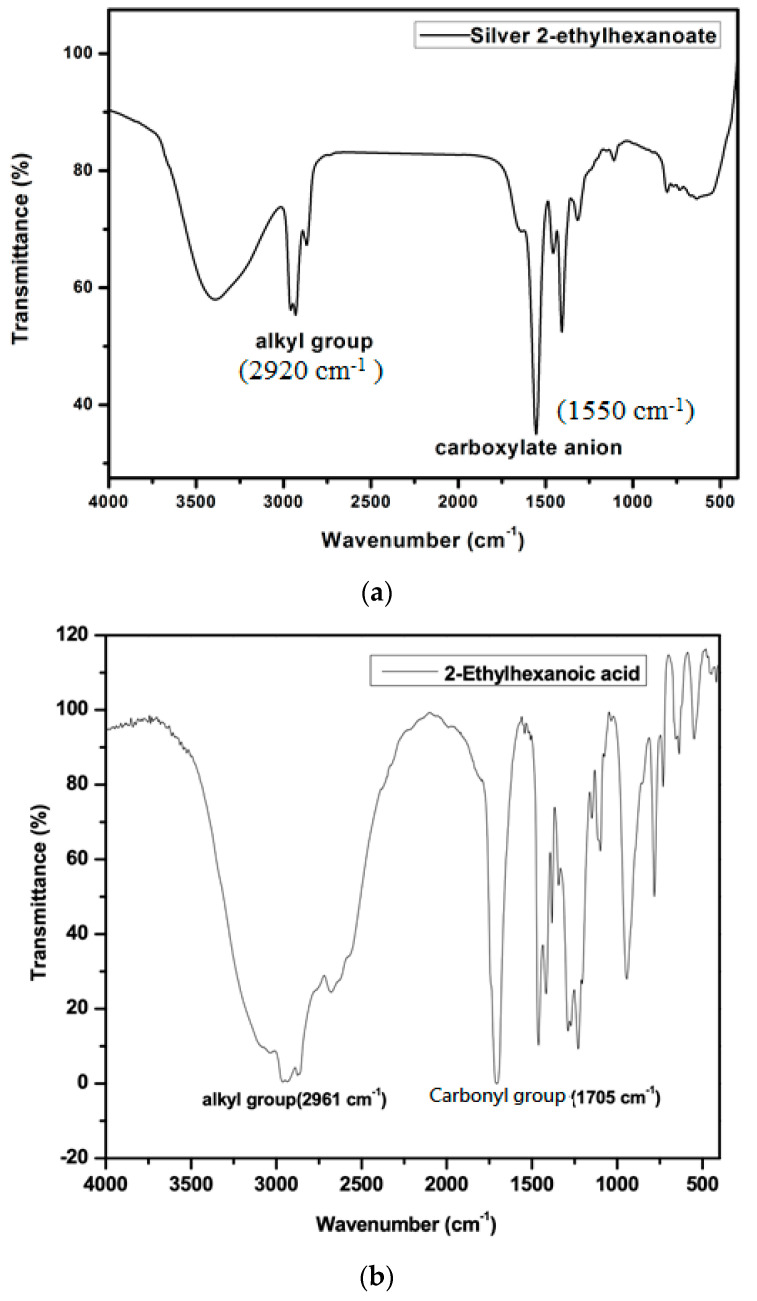
FTIR spectra of silver 2-ethylhexanoate (**a**) and 2-ethylhexanoic acid (**b**).

**Figure 5 materials-14-05941-f005:**
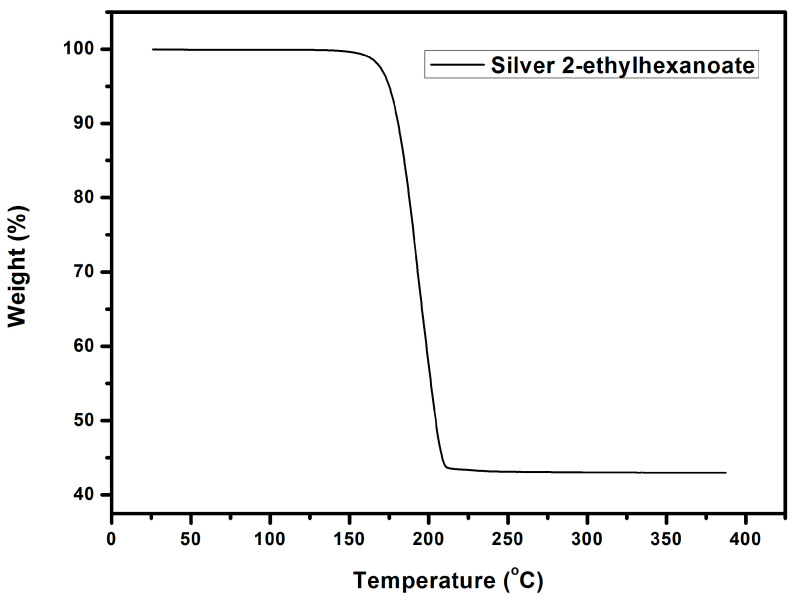
TGA of silver 2-ethylhexanoate.

**Figure 6 materials-14-05941-f006:**
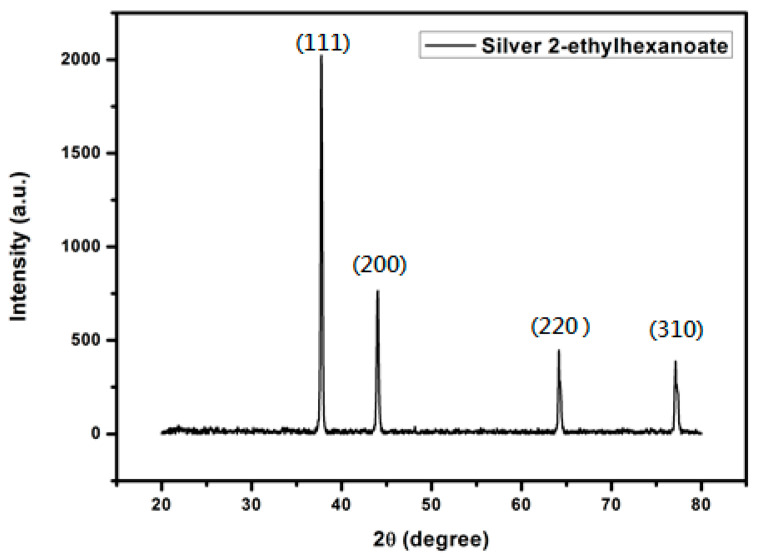
XRD pattern of silver prepared from silver 2-ethylhexanoate.

**Figure 7 materials-14-05941-f007:**
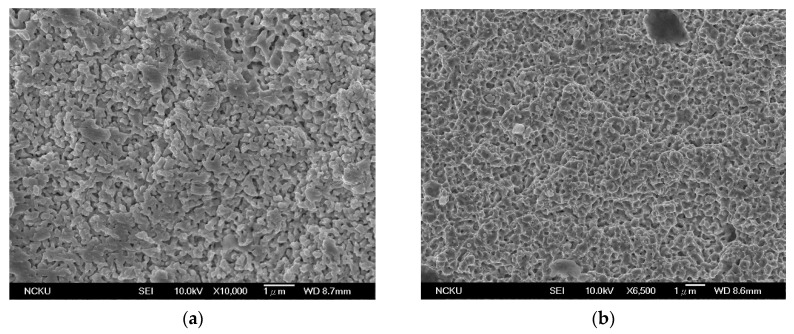
The SEM images of the fracture surfaces with different amounts of silver 2-ethylhexanoate: (**a**) PM03-0; (**b**) PM03-10; (**c**) PM03-17.5; (**d**) PM03-25.

**Figure 8 materials-14-05941-f008:**
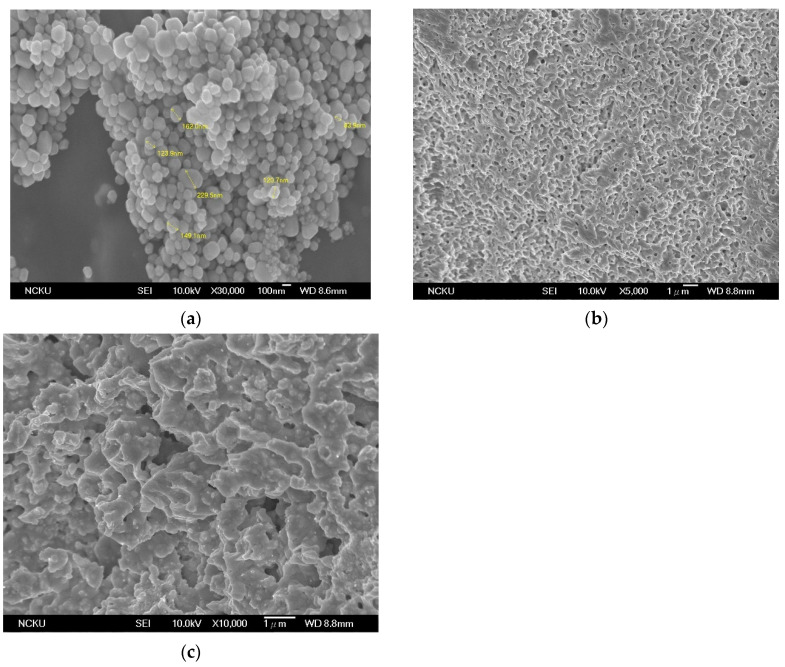
SEM images of paste PM03-17.5 sintering at different temperatures: (**a**) 200 °C; (**b**) 250 °C; (**c**) 300 °C.

**Table 1 materials-14-05941-t001:** Composition of paste PM03.

	Silver Nanoparticles	Silver 2-Ethylhexanoate	DMAc
Paste PM03-0	70 wt%	0 wt%	30 wt%
Paste PM03-10	60 wt%	10 wt%	30 wt%
Paste PM03-17.5	52.5 wt%	17.5 wt%	30 wt%
Paste PM03-25	45 wt%	25 wt%	30 wt%

**Table 2 materials-14-05941-t002:** Resistivities of the pastes with different amounts of silver 2-ethylhexanoate.

Pastes	Heating Temperature/Time	Resistivity (Ω∙m)
Paste PM03-0	250 °C/30 min	(5.86 ± 1.65) × 10^−6^
Paste PM03-10	250 °C/30 min	(1.67 ± 0.10) × 10^−6^
Paste PM03-17.5	250 °C/30 min	(3.50 ± 0.02) × 10^−7^
Paste PM03-25	250 °C/30 min	(4.97 ± 0.47) × 10^−7^

**Table 3 materials-14-05941-t003:** Shear strengths of the pastes with different amounts of silver 2-ethylhexanoate.

Pastes	Heating Temperature/Time/Applied Pressure	Shear Strength (MPa)
Paste PM03-0	250 °C/30 min/10 MPa	22.54
Paste PM03-10	250 °C/30 min/10 MPa	30.00
Paste PM03-17.5	250 °C/30 min/10 MPa	57.48
Paste PM03-25	250 °C/30 min/10 MPa	39.52

**Table 4 materials-14-05941-t004:** Resistivities of the pastes at different sintering temperatures.

Pastes	Heating Temperature/Time	Resistivity (Ω∙m)
Paste PM03-17.5	200 °C/30 min	(9.65 ± 1.13) × 10^−6^
Paste PM03-17.5	250 °C/30 min	(3.50 ± 0.02) × 10^−7^
Paste PM03-17.5	300 °C/30 min	(3.39 ± 0.24) × 10^−7^

**Table 5 materials-14-05941-t005:** Shear strengths of the pastes at different sintering temperatures.

Pastes	Heating Temperature/Time/Applied Pressure	Shear Strength (MPa)
Paste PM03-17.5	200 °C/30 min/10 MPa	7.01
Paste PM03-17.5	250 °C/30 min/10 MPa	57.48
Paste PM03-17.5	300 °C/30 min/10 MPa	60.40

**Table 6 materials-14-05941-t006:** The comparison of PM03-17.5 with other the state-of-the-art silver pastes.

Pastes [Reference Number]	Sintering Temperature (°C)	Shear Strength (MPa)	Resistivity (Ω∙m)
1 [31]	265	53	-
2 [32]	250	21	9.85 × 10^–8^
3 [33]	250	30	-
4 [34]	225	46.4	-
5 [35]	280	25	-
6 [36]	220	43.6	-
7 [37]	275	32.7	1.04 × 10^−7^
PM03-17.5 (This study)	250	57.48	3.50 × 10^−7^

## Data Availability

Not applicable.

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
