# Peer review of "Low Sintering Temperature Nano-Silver Pastes with High Bonding Strength by Adding Silver 2-Ethylhexanoate"

_materials, 2021, doi:10.3390/ma14205941_

Round 1

Reviewer 1 Report

Hsu et. al reported work "Low Sintering Temperature Nano-Silver Pastes with High Bonding Strength need to be revised before it gets published in Materials. 

1) Introduction authors need to be discuss briefly in relative comparison with reported works silver pastes effects towards bonding strengths particularly based on chemical aspects. It would be better for clear understanding. 

2) Some of SEM figures scale bars and other information is not clearly visible. 

3) FTIR spectra appropriate notations need to keep in figures like carboxylate peak (it is a general term) near 1550 cm-1, it need to be specified with particular absorption band with full functional group in the parentheses. 

4) Experimental section need to be presented in standard format with scale not simple molar representation. 

5) XRD patterns specific lattices need to be notified in the figure. 

Reviewer 2 Report

Review of paper titled Low sintering temperature nano-silver pastes with high bonding strength by S. Lien-Chung Hsu et al.

This paper studied the effects of silver 2-ethylhexanoate additions on microstructure and physical properties of silver nanoparticles. The authors found that 17.5 wt % of silver 2-ethylhexaonate provided the best result in terms of porosity reduction, shear strength increase and low electrical resistivity. The paper is well-written and materials are properly characterized. Nevertheless, the manuscript lacks a proper discussion and comparison with literature. The authors also need to highlight the novelty.

1.Silver 2-ethylhexanoate is a well-known metal-organic compound that facilitates the consolidation of silver nanoparticles (http://doi.org/10.1143/JJAP.45.6987, http://doi.org/10.1143/JJAP.46.251). Therefore, the authors need to properly highlight what exactly is new in their study.

2.The physical properties of the prepared silver nanoparticles (shear strength, porosity, electrical resistivity) have to be compared with previous results.

3.The article title should be specific enough and include the name of the precursor (silver 2-ethylhexanoate). The chemical formula of the precursor should be also given in the paper for the sake of illustration.

Reviewer 3 Report

Steve Lien-Chung Hsu and co-workers have illustrated the optimization and mixing of silver 2-ethylhexanoate and silver nanoparticles along with other variables to obtain the best performance of the nano-silver paste. However, in-depth studies should be done and this article can be considered for publications after following major modifications.

  1. Authors need to compare the FT-IR spectrum of silver 2-ethylhexanoate with 2-ethylhexanoic acid to determine the interactions.
  2. They should show the variation in FT-IR and powder XRD pattern before and after heating at 250oC for 30 min shown in Figure 4 and 6.
  3. Although SEM images have been recorded for different pastes, but atomic force microscopy (AFM) would be better to see the surface characterization.
  4. The scale should be mentioned in SEM images figure captions 7 and 8. Moreover, in Figure 8, all SEM images should be shown at the same scale.

Reviewer 4 Report

This manuscript deals with the fabrication of a new nano-silver paste. The paste with a 52.5/17.5 % wt ratio has low resisitivity ((3.50 ± 0.02) x10-7 Ω∙m) with relative high shear strength (57.48 MPa). The manuscript is concise, the paste is well characterized but there are still some points that need to be addressed:

  • The Introduction is short and needs to be expanded
  • What is the PDF associated with the XRD profile?
  • In Figure 4 the FTIR spectrum contains more peaks. Bearing in mind that for the fabrication of the paste DMAc was used, is there any possibility that organic molecules were adsorbed on the surface?
  • The resolution of the SEM images needs to be improved as the scale is not visible
  • Have the authors performed any Raman measurements?
  • Can the authors provide any proof for the final elemental composition?
  • Additionally, this manuscript needs more evidence on what makes the newly synthesized silver paste better than current commercial ones? Maybe a comparison table with more elaboration?

I would be more than happy to suggest this manuscript for publicaton in Materials after these issues have been addressed

Round 2

Reviewer 1 Report

Revised form of article entitled on " Low Sintering Temperature Nano-Silver Pastes with High Bonding Strength" can be acceptable for the publication in Materials. 

Author Response

Thank you very much for reviewing my manuscript.

Reviewer 3 Report

Authors have answered all the queries precisely well and the manuscript can be acceptable for publication.

Author Response

(The authors gave the same response as above.)

Reviewer 4 Report

The authors have replied to almost every suggestion and the quality of the manuscript has improved. I recommend the publication of this manuscript at its present form

Author Response

(The authors gave the same response as above.)
